# Psychosocial Management Before, During, and After Emergencies and Disasters—Results from the Kobe Expert Meeting

**DOI:** 10.3390/ijerph16081309

**Published:** 2019-04-12

**Authors:** Mélissa Généreux, Philip J. Schluter, Sho Takahashi, Shiori Usami, Sonoe Mashino, Ryoma Kayano, Yoshiharu Kim

**Affiliations:** 1Sherbrooke Hospital University Centre, Eastern Townships Integrated University Centre in Health and Social Services, Sherbrooke, QC J1J 3H5, Canada; 2Department of Community Health Sciences, Faculty of Medicine and Health Sciences, Université de Sherbrooke, Sherbrooke, QC J1G 1B1, Canada; 3School of Health Sciences, University of Canterbury—Te Whare Wānanga o Waitaha, Christchurch 8140, New Zealand; philip.schluter@canterbury.ac.nz; 4Primary Care Clinical Unit, School of Clinical Medicine, The University of Queensland, Brisbane QLD 4006, Australia; 5Department of Disaster Psychiatry, University of Tsukuba, Tsukuba 305-8575, Japan; takahashi.sho.fn@u.tsukuba.ac.jp; 6Ibaraki Prefectural Medical Center of Psychiatry, Kasama 309-1717, Japan; 7Department of Mental Health and Psychiatric Nursing, Faculty of Life Sciences, Kumamoto University, Kumamoto 860-8555, Japan; susami@kumamoto-u.ac.jp; 8Research Institute of Nursing Care for People and Community, University of Hyōgo, Akashi 673-8588, Japan; sonoe_mashino@cnas.u-hyogo.ac.jp; 9World Health Organization Centre for Health Development, Kobe 651-0073, Japan; kayanor@who.int; 10National Institute of Mental Health, National Center for Neurology and Psychiatry, Kodaira 187-0031, Japan; kim@ncnp.go.jp

**Keywords:** health emergency and disaster risk management (Health-EDRM), Sendai Framework for Disaster Risk Reduction, WHO Thematic Platform for Health-EDRM Research Network, post-traumatic stress disorder, mental health impacts, psychosocial management, community resilience

## Abstract

Emergencies and disasters typically affect entire communities, cause substantial losses and disruption, and result in a significant and persistent mental health burden. There is currently a paucity of evidence on safe and effective individual- and community-level strategies for improving mental health before, during, and after such events. In October 2018, the World Health Organization (WHO) Centre for Health Development (WHO Kobe Centre) convened a meeting bringing together leading Asia Pacific and international disaster research experts. The expert meeting identified key research needs in five major areas, one being “Psychosocial management before, during, and after emergencies and disasters”. Experts for this research area identified critical gaps in observational research (i.e., the monitoring of long-term psychological consequences) and interventional research (i.e., the development and evaluation of individual- and community-level interventions). Three key research issues were identified. First, experts underscored the need for a standardized and psychometrically robust instrument that classified the mental health/psychosocial risk of people within both a clinical and community setting. Then, the need for a standardization of methods for prevention, screening, diagnosis, and treatment for affected people was highlighted. Finally, experts called for a better identification of before, during, and after emergency or disaster assets associated with greater community resilience.

## 1. Introduction

Large-scale emergency and disaster events, resulting from natural, technological, or human causes, often affect entire communities. They invariably yield significant human and capital injuries and losses coupled with prolonged social and infrastructure disruption. Moreover, emergencies and disasters typically place a significant and persistent mental health burden on those directly and indirectly affected, together with those who respond and their associated services. As a result, public health organizations are charged with intervening and helping citizens and communities cope in the aftermath of emergencies and disasters. The public health response during and after emergencies and disasters has historically focused on protecting populations from immediate threats [1]. However, the role of public health and health systems in providing sustained psychosocial support for directly and indirectly affected victims, the long-term monitoring of psychological profiles, and delivering ongoing strategies for enhancing resilience may be just as important [2].

In October 2018, the World Health Organization (WHO) Centre for Health Development (WHO Kobe Centre) convened a meeting in order to identify key emergency and disaster research needs. A panel of invited experts from the WHO, WHO Thematic Platform for Health Emergency and Disaster Risk Management (Health-EDRM) Research Network (TPRN), World Association for Disaster and Emergency Medicine, and the Japan International Cooperation Agency, along with delegates to the Asia Pacific Conference for Disaster Medicine 2018, was convened [3]. The panel was informed by multiple sources, including the progress and implementation of the Sendai Framework on Disaster Risk Reduction 2015–2030 [4], documentation from the 3rd UN World Conference on Disaster Risk Reduction (including the establishment of TPRN), and literature on the recommended Health-EDRM research activities [5,6,7,8]. After a review of these documents, together with appraising existing projects and activities, five major areas of research need were identified. These were clustered and named under the following banners: (i) Health data management before, during, and after emergencies and disasters; (ii) Psychosocial management before, during, and after emergencies and disasters, and other medium- and long-term effects on the public health and health systems, the topic of this paper; (iii) Community emergency and disaster risk management, including risk literacy and addressing needs of sub-populations; (iv) Health workforce development for health emergency and disaster risk management; and, (v) Research methods and ethics. 

## 2. Material and Methods

The content of this discussion paper arises from the WHO Kobe Centre convened panel’s deliberations, based on available evidence and their expertise in psychosocial management before, during, and after emergencies and disasters. The lead discussant (Yoshiharu Kim), the rapporteur (Mélissa Généreux), and the other experts who participated to the discussion (*n* = 7) primarily sought to (i) identify critical gaps in scientific evidence in this major area of research, as well as (ii) discuss concrete research questions and issues, with the aim of potentially resolving these gaps through collaboration among Asian Pacific regional and global researchers and related stakeholders. Additionally, the panel aimed to assess knowledge-to-practice gaps in order to better integrate current expertise and research in this area for each disaster management phase.

## 3. Results

A consensus emerged among the expert panel that there is currently a paucity of evidence on safe and effective individual- and community-level strategies for monitoring and improving psychosocial health and building resilience before, during, and after emergencies and disasters. 

### 3.1. Critical Research Gap Identification

The discussion on critical gaps in scientific evidence led to the identification of three priorities in research, namely the monitoring of long-term psychological consequences; the need for individual-level interventions; and the need for community-level interventions. Each of these identified gaps are now described in turn.

#### 3.1.1. Monitoring of the Long-Term Psychological Consequences

In order to strengthen current psychosocial strategies to promote health and wellbeing before, during, and after emergencies and disasters, the monitoring of people’s psychological profile needs to be harmonized or standardized. Such standardized monitoring enables population patterns to be followed over time, across the phases of the emergency or disaster, and provides an empirical characterization of recovery for the whole population and various important subgroups. Moreover, it would enable national and international comparisons to be drawn and facilitate a better understanding of community-level psychosocial wellbeing. Ideally, any standardized instrument would be derived from routinely collected information, thereby reducing participant burden and also being available before, during, and after emergencies and disasters. Many studies have reported both persistent adverse outcomes (notably post-traumatic stress disorders (PTSD) or symptoms (PTSS), anxiety, and depression) and positive outcomes (e.g., post-traumatic growth, and closer sense of community) for years following exposure to natural or anthropogenic disasters. Moreover, these positive and negative effects have been observed among those directly affected by the emergency or disaster, and also among those peripherally affected, such as workers and caregivers, and the wider population. However, community-level profiles are often left unmeasured prior to emergencies or disasters, making longitudinal recovery patterns difficult to fully ascertain. This is further compounded by the fact that specific sub-groups are often disproportionally impacted, may have differential recovery periods, and that the impact may vary greatly according to exposure type and intensity [9,10,11,12,13,14,15]. Children and youth, the elderly, those socioeconomically deprived, and disabled people frequently carry this disproportionate burden. Although standardization of exposure and outcome measures is desirable and central, this should not preclude the additional use and reporting of other culturally or geographically specific measures. Various indigenous and other groups of people around the globe define health and wellbeing in different but equally valid ways, and this should be acknowledged and respected in any form of reporting or monitoring.

#### 3.1.2. Individual-Level Interventions

To better understand the psychosocial risks, vulnerabilities, and capacities following an emergency or disaster, a wide range of individual-level interventions need to be developed and evaluated, both in clinical and community-based settings. Such interventions should include, but are not limited to, self-care programs, humanitarian support, psychological first aid, prevention, screening and diagnosis tools for both health and non-health workers, and medication required for various types of sequelae mental health problems [16,17]. Preventive interventions are thought to be especially important to reduce the risk of PTSD and depression among workers who are acutely exposed [18], including first responders (e.g., fire, police, emergency medical staff). It is also critical to ensure that validated clinical interventions and tools can be shared and used more widely, through training, guidelines, repositories, and communication and dissemination strategies. Guidelines for the appropriate use of the interventions and tools must be clearly articulated, and their misuse must be avoided.

#### 3.1.3. Community-Level Interventions

In a similar vein to the above, there is a need to further develop and evaluate community-level strategies, such as psychosocial education (before emergency or disaster), addressing the unmet psychosocial needs through self-help groups and dynamic group psychotherapy (after emergency or disaster), cultivating community resilience, empowering the citizens, and mobilizing the community (before, during, and after emergency or disaster) [19,20,21,22]. An evidence base of interventions that have empirically demonstrated successful implementation and efficacy needs to be developed and established. As for individual-level interventions, there is an added need for enhanced knowledge translation strategies so that lessons learned from community-level interventions can be pooled, shared, and used on a broader basis.

### 3.2. Proposed Research Questions and Issues

Concrete research questions were proposed to meet each of the identified knowledge gaps. The first question pertained to the classification of people exposed to a disaster according to their level of psychosocial need. Currently, multiple and often inconsistent definitions are employed. It was advocated that an epidemiological tool (i.e., a standardized and psychometrically robust instrument) should be developed and validated for the classification of risk (e.g., high-risk, mid-risk, and low-risk) for people exposed to an emergency or disaster. Drawing on the social determinants of health framework, a wide range of risks and assets previously documented in the scientific literature should underpin the construction of this tool [23,24,25]. Broadly, it is likely to include the following:Before emergency or disaster: pre-existing risk and protective factors;During emergency or disaster: exposure to primary stressors, human and material losses;After emergency or disaster: exposure to secondary stressors, and access and use of local resources.

As the emergency or disaster impacts invariably extend beyond the clinical setting, the tool should be suitable for patients within that setting but also for persons within the wider community (e.g., conducting a community health survey). As such, it is likely the tool will have robust screening rather than definitive clinical exposure diagnostic properties. The validation of this tool should be performed using longitudinal and multi-centric studies, and should focus on its ability to predict various outcomes, like PTSD, other psychological symptoms (including psychological distress, anxiety, and depressive symptoms), maladaptive behaviors (including alcohol and drug abuse), self-care, quality of life, and positive outcomes (including post-traumatic growth).

The need to standardize interventions delivered to people exposed to a disaster using evidence-based best practices was also stressed. Panel members suggested the development and validation of methods to screen, diagnose, and treat people exposed to a disaster in clinical and community-based settings. Such clinical tools should frame emerging practices, such as risk screening by community members, by ensuring that they are both effective and safe, and by underlining the conditions for success of such methods.

Finally, panel members called for a better identification of what makes a community resilient, through an assessment of before, during, and after emergency or disaster assets (i.e., characteristics, strengths, and resources) that are associated with greater community resilience. Local knowledge should be considered in the same manner as scientific knowledge. Having been through a unique and informative experience, the local health workforce involved in psychosocial management must draw and share lessons in the aftermath of a disaster (in close collaboration with researchers and community members) through case studies. Guidelines for their reporting of these studies should include the following.

What were the needs and assets in the local community?How and by whom were these needs and assets addressed?What were the barriers and the success factors for sustaining resilience and recovery?

Standards (or format) to report case studies should facilitate the pooling and the sharing of such local evidence. Moreover, respective roles of health and non-health sectors need to be clarified, as community resilience absolutely requires a cross-sectoral approach. In time, these case studies could be subjected to meta-analyses in order to distill common features that transcend each unique emergency or disaster-ravaged community.

## 4. Discussion

The current research gaps and priorities in psychosocial management after emergencies and disasters identified by the expert panel are, perhaps unsurprisingly, mirrored within the literature. The lack of long-term observational and interventional research [8,26,27,28], challenges in systematically collating and delivering lessons learned from events, and difficulties in creating and sustaining effective community engagement [29,30,31] have all been described before. However, this paper is among the first that attempts to draw the various threads of these research needs together.

Although the before, during, and after phases of emergency and disaster management are all important, the panel asserted that local efforts in psychosocial management are mostly oriented toward the short-term after the event. Once the emergency or disaster is over, medical, psychosocial, and public health teams are often required to revert to and resume their regular tasks as quickly as possible. Long-term psychosocial recovery is perhaps the most challenging task because of the burden on professionals (e.g., accumulating workload) and organizational factors (e.g., weakening political commitment). As a result of this fatigue, monitoring of long-term psychological health consequences is not usually done (with a few exceptions) [25,32], much less in a standardized manner.

Another critical area that is typically overlooked in the post-event landscape is the evaluation of psychosocial interventions provided during the recovery phase of an emergency or disaster, which are typically not routinely carried out. Such evaluations would provide the necessary evidence for establishing standards and best practices [26]. Due to the often unpredictable nature of emergencies or disasters, it is often impossible to conduct randomized or prospective cohort studies that capture the before, during, and after phases of the event. Instead, within this emergency or disaster setting, interventional research should exploit apposite tailored natural experiment designs. These will often be shaped by pragmatic considerations, such as resources, expertise, and data availability, but are likely to include pre–post, interpreted time series, and change point study designs. Together with the standard cross-sectional or repeated measure studies often initiated following an event, a richer empirical evidence base will emerge. With the increasing availability of routinely collected data and greater ability for matching and tracking individuals over time, these data sources are likely to have increased utility in emergency and disaster research. With data sufficiency, the psychosocial profiles within affected communities could be monitored by measuring changes before, during, and after the emergency or disaster and comparing these patterns to unaffected control communities [33]. Finally, learning from past events should be systematized, notably through case studies, as they are likely to inform the identification and sharing of common international problems and solutions.

The importance of identifying and leveraging existing assets at the community level was also raised. Working from existing capacities was strongly valued among the panel. However, communities typically remain poorly engaged in disaster management. Indeed, often after an emergency or disaster, various sectors within the community, particularly those disproportionately affected, experience heightened barriers which contribute to further physical and social exclusion, and pursuant disengagement. Evidence-based efficacious strategies to mobilize local knowledge and expertise before, during, and after a disaster are yet to be developed [34].

Finally, the discussion of our expert panel on solutions to fill psychosocial management knowledge and knowledge-to-practice gaps before, during, and after emergencies and disasters aligned with the priorities set by the 2015–2030 Sendai Framework for Disaster Risk Reduction [35]. Indeed, the Sendai Framework emphasizes the requirement for *“improving the knowledge of government representatives at all levels through the sharing of experiences, lessons learned and good practices”* (Paragraph 24g) and “*ensuring that local knowledge and practices complete, as appropriate, scientific knowledge of disaster risk assessment”* (Paragraph 24i) [4].

## 5. Conclusions

It is hoped that this delineation and discussion on key research issues in psychosocial management will contribute to the identification and implementation of concrete solutions that foster the creation and the use of knowledge for psychosocial support and resilience building before, during, and after emergencies and disasters. As for other major Health-EDRM research areas, psychosocial management requires collaboration between experts, decision makers, practitioners, and communities in order to facilitate coordinated responses when and where they are most needed. It is critical that psychosocial strategies developed to support victims and communities affected by emergencies and disasters are validated and that knowledge gained from these experiences is shared and used as much as possible.

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
