# Peer review of "Psychosocial Management Before, During, and After Emergencies and Disasters—Results from the Kobe Expert Meeting"

_ijerph, 2019, doi:10.3390/ijerph16081309_

Reviewer 1 Report

This is a potentially interesting article on the psychosocial management before, during, and after emergencies and disasters based on a convening of experts at a conference on disaster medicine. The manuscript needs to be edited for English language usage and this issue makes it difficult to fully assess the manuscript for publication. As it is now, the manuscript is very unclear.

The introduction needs to be clearer and more background information is needed to contextualize this issue. The meeting was part of a conference on disaster medicine and more detail is needed on psychosocial management in this context, including what constitutes a disaster in this context. In the material and methods section, the number of participants should be noted. Another issue is that the lack of clarity in the gaps in knowledge on this issue and so the suggestions come across as broad strokes and not situated in the context of previous research conducted in this area.

The manuscript highlights an important issue. However, addressing the English language usage issue and addressing the clarity issues would allow for a better assessment of this manuscript than I can provide at this time.   

Author Response

This is a potentially interesting article on the psychosocial management before, during, and after emergencies and disasters based on a convening of experts at a conference on disaster medicine. The manuscript needs to be edited for English language usage and this issue makes it difficult to fully assess the manuscript for publication. As it is now, the manuscript is very unclear.

 The manuscript has undergone extensive English editing, through a professional English editing service.

 The introduction needs to be clearer and more background information is needed to contextualize this issue.

 More background information on disasters and their impacts on psychological well-being has been added in the Introduction section.

 The meeting was part of a conference on disaster medicine and more detail is needed on psychosocial management in this context, including what constitutes a disaster in this context.

 In the Introduction section, disasters have been better described et more details have been provided on the importance of strengthening psychosocial management strategies in this context.

 In the material and methods section, the number of participants should be noted.

 The number of participants has been added as suggested in the Material and methods section.

 Another issue is that the lack of clarity in the gaps in knowledge on this issue and so the suggestions come across as broad strokes and not situated in the context of previous research conducted in this area.

 The Result sections now starts with a clear statement on the main gaps in knowledge on this issue that were identified by the participants.

 We have also added more background information in the Introduction section so that the context of this research area is better understood.

 The manuscript highlights an important issue. However, addressing the English language usage issue and addressing the clarity issues would allow for a better assessment of this manuscript than I can provide at this time.   

We have reorganized the manuscript so that :

- the Introduction focuses on a) background information to contextualize the issue of psychosocial management before, during and post-disasters, b) the expert meeting held in October 2018, and c) the aim of the paper ;

- the Material and methods section, which has been shortened,  focuses on the experts that participated to the meeting and the points of discussion that were addressed by these experts on the major research area “Psychosocial management before, during and after emergencies and disasters” ;

- the Result section focuses on critical gaps in scientific evidence in this major area of research, and on concrete research questions and issues to fill these gaps, that were identified by the experts during the meeting and supported by the scientific literature ;

- The Discussion section is unchanged.

Thank you for your comments. We believe the revised version of the manuscript is significantly improved.

Reviewer 2 Report

Proofreading and editing is suggested of the final version as there are some grammatical errors in the document, e.g., lines 53-58. This sentence is unclear and far too long. I suggest revising into shorter sentences and clarifying the meaning.

In the abstract the abbreviation is Health-EDRM, but in the article it is H-EDRM. I advise being consistent.

The abstract describes the developments that led to the identification of the topic of this paper, but this should be in the introduction. The detail in the abstract need to be scaled back and the gist of it presented in the paper.

I find the paper hard to read and not well structured. In my opinion, the content of the different sections are not well organised. For instance some of the text in the Methods section seems to me to belong to the Introduction, similarly bits of the results section need to be in the method section.

I suggest a major edit of the article with a better organisation of content and a clearer description of the aims of the paper.

Author Response

Proofreading and editing is suggested of the final version as there are some grammatical errors in the document, e.g., lines 53-58. This sentence is unclear and far too long. I suggest revising into shorter sentences and clarifying the meaning.

This sentence (lines 53-58) has been revised into shorter sentences and its meaning has been clarified, as suggested.

The manuscript will undergo extensive English editing, through a professional English editing service.

In the abstract the abbreviation is Health-EDRM, but in the article it is H-EDRM. I advise being consistent.

The abbreviation H-EDRM is now used consistently.

The abstract describes the developments that led to the identification of the topic of this paper, but this should be in the introduction. The detail in the abstract need to be scaled back and the gist of it presented in the paper.

We have reorganized the manuscript so that :

- the Introduction focuses on a) background information to contextualize the issue of psychosocial management before, during and post-disasters, b) the expert meeting held in October 2018, and c) the aim of the paper ;

- the Material and methods section, which has been shortened,  focuses on the experts that participated to the meeting and the points of discussion that were addressed by these experts on the major research area “Psychosocial management before, during and after emergencies and disasters” ;

- the Result section focuses on critical gaps in scientific evidence in this major area of research, and on concrete research questions and issues to fill these gaps, that were identified by the experts during the meeting and supported by the scientific literature ;

- the Discussion section is unchanged.

I find the paper hard to read and not well structured. In my opinion, the content of the different sections are not well organised. For instance some of the text in the Methods section seems to me to belong to the Introduction, similarly bits of the results section need to be in the method section.

Please see the answer above that describes how the manuscript has been reorganized to take into account your comments.

I suggest a major edit of the article with a better organisation of content and a clearer description of the aims of the paper.

The aim of the paper is now better described at the end of the Introduction section.

Thank you for your comments. We believe the revised version of the manuscript is significantly improved.

Reviewer 3 Report

This was a content expert discussion, not research

The summary of the discussion is very good, but not research findings

The creation of research questions to create a tool for evaluating pre- peri- and post-disaster psychological maladies is very good for a study in the future

Suggest using pre, peri and post OR before, during or post-disaster (preferred)

Avoid overuse of "etc."

This is an essential aspect of disaster preparedness and response with global implications; with some modification, I look forward to reading the final publication

Author Response

This was a content expert discussion, not research.

Yes, this is right. This is now clearly specified at the beginning of the Material and methods section.

The summary of the discussion is very good, but not research findings.

We have reorganized the manuscript so that :

- the Introduction focuses on a) background information to contextualize the issue of psychosocial management before, during and post-disasters, b) the expert meeting held in October 2018, and c) the aim of the paper ;

- the Material and methods section, which has been shortened,  focuses on the experts that participated to the meeting and the points of discussion that were addressed by these experts on the major research area “Psychosocial management before, during and after emergencies and disasters” ;

- the Result section focuses on critical gaps in scientific evidence in this major area of research, and on concrete research questions and issues to fill these gaps, that were identified by the experts during the meeting and supported by the scientific literature ;

- the Discussion section is unchanged.

The creation of research questions to create a tool for evaluating pre- peri- and post-disaster psychological maladies is very good for a study in the future.

Thank you.

Suggest using pre, peri and post OR before, during or post-disaster (preferred).

The terms before, during and post-disaster are now used consistently.

Avoid overuse of "etc."

The overuse of “etc.” has been corrected.

This is an essential aspect of disaster preparedness and response with global implications; with some modification, I look forward to reading the final publication

Thank you for your comments. We believe the revised version of the manuscript is significantly improved.

Round  2

Reviewer 1 Report

There are still English language usage issues that make this article difficult to read. In general, it is not a very clear article. The English language usage issue and the lack of clarity makes it difficult to fully assess this article. Additionally, consistent language should be used for before, during, and after disasters. Stick to the terms used in the title. 

Author Response

The entire manuscript has been substantially revised by two of the co-authors, so that it is now much more easier to read.

Round  3

Reviewer 1 Report

The overall is fine.